# Using Social Media to Assess Expressions of Gratitude to God: Issues for Consideration

**Louis Tay** [1,*] , **Stuti Thapa** [1] , **David B. Newman** [2] **and Munmun De Choudhury** [3]

1   Department of Psychological Sciences, Purdue University, West Lafayette, IN 47907, USA
2   Department of Psychiatry and Behavioral Sciences, University of California, San Francisco, CA 94107, USA
3   School of Interactive Computing, Georgia Institute of Technology, Atlanta, GA 30332, USA
*   Correspondence: stay@purdue.edu

**Abstract:** With the proliferation of technology-based communication, public expressions of gratitude to God on social media have become more pervasive. At the same time, data science approaches are increasingly being applied to social media language data to assess positive human attributes. We elucidate critical considerations in assessing public expressions of gratitude to God, including language variability and comparability, degree of authenticity, machine learning language analysis, and aggregation approaches that could affect assessment accuracy.

**Keywords:** gratitude expressions; public gratitude; social media; data science; machine learning; challenges; review

## 1. Introduction

Expressing gratitude to God is deeply rooted in many different religious traditions that call for adherents to develop and cultivate it as moral virtue (Emmons and Crumpler 2000). Recent work also points to how gratitude to God is associated with well-being outcomes such as better health, greater subjective well-being, and lower depressive symptoms (Aghababaei and Tabik 2013; Krause et al. 2014; Krause et al. 2015; Rosmarin et al. 2011).

Research on gratitude to God has been built upon the self-report survey paradigm (e.g., Krause 2006; McCullough et al. 2002), which has unique strengths such as allowing individuals to report on their own experiences and intentions. The administration of self-report questionnaires is fairly routine, and many participants are familiar with this method. Moreover, self-report assessments provide researchers with the flexibility and ease to examine precise measures and specific psychological mechanisms. Because the tradition of self-report has existed for many years, numerous resources that provide recommendations and guidance on construct validity exist (Cronbach and Meehl 1955; Loevinger 1957; Messick 1980; Messick 1995; Cronbach and Meehl 1955; Loevinger 1957; Messick 1980, 1995). Self-report methods can also be used in a variety of different methods, such as experiments, cross-sectional studies, longitudinal designs, and daily diary and ecological momentary assessment methods.

Nevertheless, self-report surveys have several weaknesses. For example, they do not provide observable expressions of gratitude. As aligned with the Theory of Planned Behavior (Ajzen 1991), intentions to express gratitude do not always veridically emerge as gratitude expression behaviors. More broadly, self-ratings have multiple problems stemming from a lack of self-awareness, misremembering, or the use of heuristics that bias their accuracy (Schwarz 1999). Desirable responding may also influence participants' responses, resulting in reports that may not match their true beliefs or behaviors (Nederhof 1985). These examples highlight a broader principle, namely that all methods are flawed in various manners and have their own strengths and weaknesses (McGrath 1981). Therefore, the best approach in moving gratitude to God research forward is to rely on various, potentially complementary methods and acknowledge their own strengths and weaknesses.

One method that can be used to complement self-report methods is the use of text analytic techniques of social media language. This appears to be a promising candidate because social media applications are widely used by people across the world. It is estimated that 4.48 billion people worldwide use social media as of 2021 (Deen 2021). Importantly, from a cursory glance at social media posts, people often express gratitude to God on these platforms (e.g., 'thank god', 'praise god'), and these instances provide directly observable behaviors that happen in a naturalistic setting without researcher solicitation. Our team applied search terms (20 key terms, 10 God-focused and 10 interpersonal/general) to find relevant tweets, which resulted in a large quantity of observable behavior of gratitude towards God. Limiting it to the year 2019–2020, we found 1.2 million Tweets referencing gratitude; of the subsample of 105 k Tweets that were initially studied, 29.3% of the Tweets referenced gratitude toward God. We note that researchers more generally can use existing libraries such as the Linguistic Inquiry and Word Count (LIWC; Pennebaker et al. 2015) to create a gratitude dictionary such as gratef*, grati*, thank*, or appreciat* (the "*" symbol means that all words that start with the same letters are included) (e.g., Anicich et al. 2022) and also consider specific gratitude terms (e.g., praying hands emoji, tgif). Such an approach would still need domain-specific validation, as dictionaries do not necessarily capture nuances in language. A natural language processing-based approach on the other hand is able to capture and learn from linguistic patterns (e.g., sequences of words), thus with the potential to provide better validity to gratitude measurements. For readers interested in text analytics in general, and how they can apply it to assess psychological phenomena via social media, we refer readers to reviews by Eichstaedt et al. (2021) and Tay et al. (2020).

Another strength is that the application of machine learning text analysis can provide not only individual but also communal assessments of gratitude to God in a potentially scalable, longitudinal, and cost-effective fashion. Achieving accurate assessments enables measurements at the levels of communities and geographic regions. Indeed, it has been proposed that gratitude can serve as a moral "barometer" for society (McCullough and Tsang 2004) and can serve to motivate prosocial behavior (Grant and Gino 2010) and civic engagement (Panagopoulos 2011), which are vital for communities. Further, this opens up possibilities to investigate antecedents and outcomes at different levels of analysis, such as determining the national conditions that predict expressions of gratitude to God, and how these expressions may evolve over time when punctuated by local or global events (such as the Coronavirus pandemic).

While there are strengths to using social media to assess expressions of gratitude to God, there are also critical considerations to ensure its validity and reliability. This paper elucidates these issues so that researchers can identify potential pitfalls and amelioration strategies when applying this approach.

## 2. Language Variability and Comparability

Given social media usage across different cultures and nations, there is an opportunity to capture different instantiations of gratitude to God expressions. On the linguistic front, they include formal or informal language (González Bermúdez 2015; Utami et al. 2019), the use of non-standard abbreviations, short forms, deliberate misspellings and grammar (Eisenstein 2013), the use of dialects (Huang et al. 2016), the use of emojis (Guntuku et al. 2019), the use of slang and colloquialisms (Reyes et al. 2012), the use of word lengthenings (Brody and Diakopoulos 2011), and the use of different types of languages (Li et al. 2020). Multi-language and cross-cultural investigations (e.g., De Choudhury et al. 2017) can inform us of differences in gratitude expressions that need to be taken into consideration, along with similarities in patterns of expressions (e.g., Guntuku et al. 2019; Li et al. 2020). This is made more complex by the different perspectives that religions hold on the concept of the divine: there is wide variation in the "God" worshipped (e.g., Allah, Christ, etc.). Similarly, the nature of gratitude can also vary based on language norms within a religion (e.g., "it is a blessing"; "praise God"). For instance, in Arabic, word pairs like "ALHAMD

LELLAH" which means thanks God and "ALLAH AKBAR" which means God is the greatest are used to express gratitude to God, but some also write them as a single word (Rabie and Sturm 2014). Cultural knowledge is necessary to understand these language norms and variations to properly measure them for cross-cultural studies. Clearly, while language on social media is not constrained like self-report ratings, researchers have to work through a vast multiplicity of gratitude expressions. These need to be carefully curated based on experience and expertise of the religion, culture, and language. Some examples include forming a cross-cultural team of researchers (Tam and Milfont 2020) who are involved in the community, use of qualitative or survey responses to understand context-specific knowledge (Broesch et al. 2020), as well as incorporating mixed methods that allow researchers to measure and compare these cross-cultural differences (Schrauf 2018).

The breadth of possibilities also raises the question of comparability across languages, cultures, and religions. At one level, a frequency approach, or counting the number of instances (or posts) where gratitude to God is expressed, may appear to have fewer issues in terms of comparability, providing the threshold for what counts as gratitude to God. At another level, an intensity approach (e.g., Madisetty and Desarkar 2017), where one seeks to capture the intensity of gratitude to God across expressions (e.g., "Thank God!!!!" Versus "Praise God") can be more challenging. This is compounded by the possibility of different languages, dialects, and the like. There are also additional issues in how aggregation is done, as discussed later.

In general, we recommend that the research team should set parameters around what language, region, nation, culture, and religion are being examined and obtain the appropriate expertise to determine what expressions are typically considered gratitude to God. Another possibility is to recruit active social media users who describe different ways in which they express gratitude to God, which provides researchers with exemplars of social media expressions.

## 3. Authenticity of Expressions

As expressions of gratitude to God on many social media platforms are public, it may be difficult to discern whether these expressions are genuine. Performative use of social media, in Erving Goffman's terms (Goffman 1959), has been observed in multiple contexts, and on these platforms, people are known to self-present or self-enhance (Hogan 2010), which would question their authenticity. For example, people may express mere rhetoric to look good in the eyes of other religious individuals. In addition, some expressions of gratitude may be subtle attempts to boast about themselves (e.g., "I truly can't explain why, but I find beauty in the ugliest/darkest things. #ThankYouGod") or pretenses to celebrate positive news (e.g., "I won this award for the 4th time running!! God is always on my side and I cannot thank His blessings enough") rather than sincere gratitude to God.

By extension, a lack of such expressions may reflect unease in publicly expressing gratitude to God. In fact, many religions instruct people to express gratitude to God through private prayer, which means that many expressions of gratitude to God may not be found on social media posts. Due to this, expressions of gratitude to God that exist on public social media may not be representative of the typical expressions of gratitude to God that occur naturally in daily life. That is, they may lack ecological validity. Nevertheless, the expressions of gratitude to God on public social media may still hold great value in predicting other types of experiences that may occur exclusively on such platforms. The idea is that this is the new reality in which people communicate and experience life and so expressions of gratitude to God measured through social media can predict outcomes such as expressions of well-being or engagement on social media.

Similarly, posts that reference gratitude to God also need to be differentiated from actual gratitude behavior. This can include things like advice for others (e.g., "Thanking your close ones every day is necessary. Appreciate people you have in your life!"), scripture quotes ("Rejoice evermore. Pray without ceasing. In everything give thanks: for this is the will of God in Christ Jesus concerning you. 1 Thessalonians 5: 16–18"), or gratitude

additions to unrelated posts ("#selfie #GodIsGood"). Such considerations can be made in the annotation phase so that gratitude detectors or classifiers can also make such distinctions (more below).

Social media is also a place where people tend to engage in trolling behavior in which individuals seek to provoke others through their posts (Hannan 2018). There are good possibilities that gratitude to God expressions are in the context of trolling or, more broadly, inauthentic expressions. For example, depending on the context, audience, and political ideology of the expresser, a post such as "thank God for Trump!" or "praise God for Biden" may be insincere and sarcastic. Given this nuance, there needs to be additional care to examine the context of the language used beyond simple word counts (e.g., the number of times "thank God" is used). We note that the detection of sarcasm, irony, humor, and flippant remarks on social media is challenging, and it is an active area of research (Joshi et al. 2018).

On this issue of authenticity, we encourage researchers to be mindful of the limitations of counting every expression of gratitude to God on social media as genuine heartfelt expressions. There are likely differences in authenticity based on the normative culture of expressing gratitude to God online and the social context of the expression. More generally, expressions of gratitude to God on social media may not accurately capture a person's internal states, although they may still be very useful for assessing *perceptions* and *reactions* of others witnessing these gratitude expressions; or to understand why people express gratitude online publicly. In short, gratitude expressions online and self-reported gratitude may have only some degree of overlap and not have high convergence, but they could have good divergent validities in predicting different outcomes.

## 4. Machine Learning Process

One popular process for distilling information or inferring latent attributes and behaviors from social media language is to use machine learning. For instance, such methods can enable automatically identifying and/or assessing the rate or the intensity of a phenomenon of interest (Kern et al. 2016). A major advantage is the ability to scale the assessment of gratitude to God expressions to millions of social media posts in an efficient manner. Nevertheless, the building of these text classifiers (i.e., an algorithm that identifies whether a post or sentence expresses gratitude to God, in this case) comes with its own set of challenges that also require careful consideration to ensure accuracy and validity.

Due to the variability of language, supervised machine learning is often used where human annotators provide the "ground truth" (Tay et al. 2020). Human annotators will typically rate posts on whether it expresses gratitude to God, which is then used to train machine learning algorithms. This requires careful training of annotators and the development of a replicable process for how collective decisions are made. For instance, one needs to provide examples and practice with feedback to ensure that they accurately classify posts that reference gratitude expression to God. It is also important to determine the extent annotators agree with one another on each post, or inter-rater agreement (e.g., Krippendorff's alpha-reliability) (Hayes and Krippendorff 2007). One also needs to be mindful that the algorithms developed may inherit the possible accuracies and biases of the annotators collectively (Tay et al. 2022). Finally, construct validity issues stemming from training data bias or dataset shift (i.e., training data for the machine learning model has a different distribution from the test data) may further paralyze the practical use of machine learning models (Ernala et al. 2019), while under-specified or opaque machine learning models may present ethical issues (Chancellor and De Choudhury 2020).

In the light of these challenges, one way that supervised machine learning can be implemented to study gratitude to God is by having social media users provide self-report ratings of the extent to which they express gratitude to God. These ratings can be treated as individual differences of gratitude to God. Using predictive modeling, the algorithms developed could then seek to predict which types of social media language reflect individual differences in gratitude to God. In this regard, the social media language

will be studied not as a display of direct expressions of gratitude to God, rather, the process will capture all the different types of words used by people who generally express gratitude to God. This can inform a qualitative understanding of how people publicly express gratitude to God.

In general, researchers need to be aware that training algorithms to detect expressions of gratitude to God is conceptually distinct, though possibly related, to training algorithms to detect personal subjective ratings of their own gratitude to God. The former emphasizes observable language behaviors, whereas the latter emphasizes the trait of gratitude to God in individuals.

## 5. Aggregation Approaches

Another key consideration in the use of social media language is how one aggregates the data to make inferences. It is possible to aggregate within an individual to assess the proportion of times an individual expresses gratitude to God (vs. not). In this regard, one needs to determine whether there needs to be a base number of posts an individual should have in order to reduce sampling biases. This is because any single post will have greater weight for people who have very few posts on a platform. For example, someone who has a single post on a platform, and if it so happens to be one expressing gratitude to God, this individual will be counted as 100% expressing gratitude to God. Similarly, principles behind the sampling biases often considered in survey research, such as selection bias, nonresponse bias, attrition, etc., (Olsen 2006) will also be relevant here; appropriate adjustments may be considered, such as weighting (based on overall number of posts) (Royal 2019) or propensity score matching (when making group-level comparisons) (Caliendo and Kopeinig 2008).

Due to the widespread use of social media, one may also seek to infer community- or geographic-level gratitude to God expressions. In this case, one needs to consider whether the aggregation will be done at the post or individual level. If done at the post level, one takes a count of the gratitude to God posts within a community or geographic region. However, because an individual can contribute to multiple posts, superusers may disproportionately be represented in such an approach. Another approach is to aggregate posts at the individual level first to obtain individuals' level of gratitude to God expressions; then, one proceeds to aggregate individual levels to the community or geographic level (Giorgi et al. 2018). While this approach limits the problem of superusers being disproportionately represented, there needs to be a sufficient number of active users on a platform to aggregate accurately. The issue of active users also raises the issue of the representativeness of a social media sample. For example, commonly used social media in text mining research such as Twitter has a skewed distribution with 42% of the userbase being ages 18–29 while 65+ only comprise 7% of the userbase (Pew Research Center 2022). Additional methods need to be adopted in this case to counter the digital divide (Van Dijk 2020), the gap between demographics at different socioeconomic levels in their access to information and communication technologies (ICT) and digital media (DiMaggio et al. 2001), as well as disproportionate levels of social media use in different communities and geographic regions.

As we have shown, the way aggregation is done from social media posts can be meaningfully different. Researchers who seek to extract social media language to index expressions of gratitude to God will need to make decisions on how best to perform aggregation. Beyond aggregation, there are also analyses that can be done to account for the structure of the data (i.e., individual posts nested within individuals which are in turn nested within communities), such as multilevel modeling (Raudenbush and Bryk 2002); this enables researchers to determine the level-specific predictors of these posts (e.g., what predicts the occurrence of the post within an individual; what predicts who tends to express gratitude to God; what communal factors predict communities that express more gratitude to God).

## 6. Conclusions

As researchers seek to move toward more observable behavioral approaches to capture gratitude to God expressions, social media language has become a prominent candidate as there are establishing machine learning techniques to harness such data. Nevertheless, there are also different issues that need to be considered when applying this source of data and the techniques associated with it. We hope that as the challenges are identified and made transparent, this will help researchers take steps to address or acknowledge limitations.

**Author Contributions:** Conceptualization, L.T., S.T., D.B.N. and M.D.C.; writing—original draft preparation, L.T.; writing—review and editing, S.T., D.B.N. and M.D.C.; funding acquisition, L.T., D.B.N. and M.D.C. All authors have read and agreed to the published version of the manuscript.

**Funding:** This research was funded by the John Templeton Foundation, grant number 61513.

**Institutional Review Board Statement:** Not applicable.

**Informed Consent Statement:** Not applicable.

**Data Availability Statement:** Not applicable.

**Conflicts of Interest:** The authors declare no conflict of interest.

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
