# Peer review of "Using Social Media to Assess Expressions of Gratitude to God: Issues for Consideration"

_religions, doi:10.3390/rel13090778_

Round 1

Reviewer 1 Report

I believe that this will be a valuable resource for researchers interested in assessing gratitude to God (GTG) using text analytics.

I would recommend only some minor revisions to framing in order to help readers orient to the literature on text analytics and better realize how these techniques may be applied to GTG. Specifically, I recommended more introductory material on text analytics: discussion of theory behind this method, how it's done in practice, and a general description of the kinds of studies that can be conducted using these techniques. I'd also like to see a brief review of how text analytics have been used for other topics (potentially those that are somewhat similar to GTG), so that researchers can get a better idea of how these techniques have been applied and what questions they are suited to answer. Finally, perhaps more could be done to illustrate how text analytics could be applied to the study of GTG; currently, there is great material on the potential and limitations of using text analytics for studying GTG, but the article is (IMO) lacking a more concrete description of how GTG could be studied using these techniques. 

Author Response

Reviewer 1

I believe that this will be a valuable resource for researchers interested in assessing gratitude to God (GTG) using text analytics.

I would recommend only some minor revisions to framing in order to help readers orient to the literature on text analytics and better realize how these techniques may be applied to GTG. Specifically, I recommended more introductory material on text analytics: discussion of theory behind this method, how it's done in practice, and a general description of the kinds of studies that can be conducted using these techniques. I'd also like to see a brief review of how text analytics have been used for other topics (potentially those that are somewhat similar to GTG), so that researchers can get a better idea of how these techniques have been applied and what questions they are suited to answer. Finally, perhaps more could be done to illustrate how text analytics could be applied to the study of GTG; currently, there is great material on the potential and limitations of using text analytics for studying GTG, but the article is (IMO) lacking a more concrete description of how GTG could be studied using these techniques. 

Thank you for this comment. We have added more information at the start to expand a little more on text analytics within social media. Specifically, we discuss the typical processes of using dictionaries to capture specific language. Although the scope of the paper focuses on gratitude to God, we now also refer readers to general reviews of social media text mining that can help readers understand and better use these techniques. See below.

Our team applied search terms (20 key terms, 10 God focused and 10 interpersonal/general) to find relevant tweets, which resulted in a large quantity of observable behavior of gratitude towards God. Limiting it to the year 2019-2020, we found 1.2 million Tweets referencing gratitude; of the subsample of 105k Tweets that were initially studied, 29.3% of the Tweets referenced gratitude toward God. We note that researchers more generally can use existing libraries such as the Linguistic Inquiry and Word Count (LIWC; Pennebaker et al., 2015) to create a gratitude disctionary such as gratef* gratef*, grati*, thank*, or appreciat* (the “*” symbol means that all words that start with the same letters are included) (e.g., Anicich et al., 2022) and also consider specific gratitude terms (e.g., praying hands emoji, tgif). Such an approach would still need domain specific validation, as dictionaries do not necessarily capture nuances in language. A natural language processing based approach on the other hand is able to capture and learn from linguistic patterns (e.g., sequences of words), thus with the potential to provide better validity to gratitude measurements. For readers interested in text analytics in general, how they can apply it to assess psychological phenomena via social media, we refer readers to reviews by Eichstaedt et al. (2021) and Tay et al. (2020).

Reviewer 2 Report

Evaluating expressions of gratitude to God (GTG) on social media are likely to be important for furthering the study of GTG. This paper offers sage advice for those who wish to pursue this approach to GTG, and thus the content appears to be very appropriate to Religionsand should also be of interest to its readership. 

General Commentary

1.     I found the paper to be very well written.

2.     I found the authors’ argument for how GTG expressions on social media to be an actual expression of gratitude to be a compelling point. 

3.     I found the discussion as to what might be inauthentic expressions of GTG on social media to be very helpful, and for me this is a critical issue for GTG research. 

4.     One of the significant issues for the use of social media for research is the fact that we are studying a biased population (younger, more technically savvy, etc.). Individual differences in terms of who uses social media and who does not, are interesting and must affect our interpretations of the social media data. Personally, I would have liked to have seen more discussion of this issue and how it might be overcome. 

In sum, for someone like myself who is not well versed in text analysis, this review provides a wonderful introduction for those who want to apply this approach to GTG. The analysis of social media posts for investigating GTG is likely to be an important approach, and this paper should provide a helpful guide for those of us who seek to study GTG in this way. 

Author Response

Evaluating expressions of gratitude to God (GTG) on social media are likely to be important for furthering the study of GTG. This paper offers sage advice for those who wish to pursue this approach to GTG, and thus the content appears to be very appropriate to Religions and should also be of interest to its readership. 

General Commentary

  1. I found the paper to be very well written.

 Thank you for your review and for your positive comment on our paper.

  1. I found the authors’ argument for how GTG expressions on social media to be an actual expression of gratitude to be a compelling point. 

 We appreciate this.

  1. I found the discussion as to what might be inauthentic expressions of GTG on social media to be very helpful, and for me this is a critical issue for GTG research. 

 Thank you.

  1. One of the significant issues for the use of social media for research is the fact that we are studying a biased population (younger, more technically savvy, etc.). Individual differences in terms of who uses social media and who does not, are interesting and must affect our interpretations of the social media data. Personally, I would have liked to have seen more discussion of this issue and how it might be overcome. 

This is a great point and we have discussed this a little more in the paper. See below.

The issue of active users also raises the issue of the representativeness of a social media sample. For example, commonly used social media in text mining research such as Twitter has a skewed distribution with 42% of the userbase being ages 18-29 while 65+ only comprise of 7% of the userbase (Pew Research, 2022). Additional methods need to be adopted in this case to counter the digital divide (Van Dijk 2020), the gap between demographics at different socioeconomic levels in their access to information and communication technologies (ICT) and digital media (DiMaggio et al., 2001), as well as disproportionate levels of social media use in different communities and geographic regions.

In sum, for someone like myself who is not well versed in text analysis, this review provides a wonderful introduction for those who want to apply this approach to GTG. The analysis of social media posts for investigating GTG is likely to be an important approach, and this paper should provide a helpful guide for those of us who seek to study GTG in this way. 

We appreciate the positive views on this research.

Reviewer 3 Report

This manuscript provides an overview of how text analysis of social media could be used to learn about gratitude to God.  The idea to use social media and text mining/analysis techniques is a good idea to tackle theoretically-relevant questions from a different angle, either to provide convergent evidence or to gain novel insights into how gratitude to God is expressed in the real world.  The authors nicely describe several of the benefits and challenges associated with analyzing text, and social-media-based text in particular.  For example, as the authors describe how these expressions of gratitude to God are real-world behavior that can be gathered in large quantities from diverse populations, and then be used to understand other (regional or temporal) group phenomena, such as responses to important events or cooperation within communities.  Clearly, text analysis offers several benefits beyond what can be garnered through self-report measures alone.  However, I have a few remaining questions about how gratitude to God could be studied using text analysis methods.

1.     I found myself wondering throughout this manuscript whether there were any benefits that were specific to understanding gratitude, or gratitude to God, that would not also be shared by studying any other topic using text and social media.  Is there something specific we can learn about gratitude to God that makes this method especially important for use on this topic, or is this a general argument for using more text analysis to answer social scientific research questions? 

2.     The authors give several examples of phrases that indicate gratitude to God which are typically found on social media platforms.  These certainly seem to match my subjective impressions of social media, but I wonder if the authors could also make their argument even more compelling by attaching some numbers to the frequency of these behaviors.  Of course a full study of online expressions of gratitude requires a different, more empirically-focused paper, but if the authors could do a quick search for the frequency of certain terms (e.g., “praise God”) it could provide even more support for their claim that this is an important real-world behavior worthy of study.

3.     On page 2 the authors argue that “Cultural knowledge is necessary to understand these language norms and variations to properly measure them for cross-cultural studies.”  They could provide further discussion of how one could gather this cultural knowledge.  Does it require preliminary self-report data with pilot samples? Ethnography? Recruiting research assistants from the local context you want to study? Consulting polls of social media users?

4.     In describing the potential issues with ecological validity or representativeness of these expressions, the authors state that “the expressions of gratitude to God on public social media may still hold great value in predicting other types of experiences that may occur exclusively on such platforms” (lines 127.128).  This sentence could use more examples of what other types of platform-based experiences might be theoretically-interesting.

5.     The authors repeatedly mention the necessity to find good focal words to count in text-type data.  Is there at least some good, validated, pre-existing dictionary of gratitude-related words?  If so, the authors could direct readers to it here.  Alternatively, the authors could be explicit that such a dictionary does not already exist, and that users would need to make their own.

6.     The authors discuss extensively the issues with potential lack of authenticity (e.g., ulterior motives, sarcasm, lack of prototypicality) in gratitude expressions on social media.  This implies that expressions on social media may be limited in their ability to accurately capture internal mental states.  It also seems to imply that this method may be better suited for testing research questions pertaining to witnessing other people’s expressions of gratitude, or why people choose to express gratitude in public forums like social media.  This may be another way that social media analyses could offer insights that are distinct from (not merely convergent with) self-report data.

7.     A definition could be provided for the term “the digital divide” (line 217).

8.     The authors could reference which specific methods can be used to do the actual text gathering and analysis, or at least reference some papers that provide more thorough and concrete guidelines on these methods.

9.     There are a few typos that can be corrected with further proofreading (e.g., the repeated citations in lines 32/33).

Author Response

This manuscript provides an overview of how text analysis of social media could be used to learn about gratitude to God.  The idea to use social media and text mining/analysis techniques is a good idea to tackle theoretically-relevant questions from a different angle, either to provide convergent evidence or to gain novel insights into how gratitude to God is expressed in the real world.  The authors nicely describe several of the benefits and challenges associated with analyzing text, and social-media-based text in particular.  For example, as the authors describe how these expressions of gratitude to God are real-world behavior that can be gathered in large quantities from diverse populations, and then be used to understand other (regional or temporal) group phenomena, such as responses to important events or cooperation within communities.  Clearly, text analysis offers several benefits beyond what can be garnered through self-report measures alone.  However, I have a few remaining questions about how gratitude to God could be studied using text analysis methods.

  1. I found myself wondering throughout this manuscript whether there were any benefits that were specific to understanding gratitude, or gratitude to God, that would not also be shared by studying any other topic using text and social media.  Is there something specific we can learn about gratitude to God that makes this method especially important for use on this topic, or is this a general argument for using more text analysis to answer social scientific research questions? 

The Introduction section of our paper describes the significance of understanding gratitude, including gratitude to God, and highlights what might be offered by approaches to assessing it via social media. That said, such computational methods can apply generally to other topics too, such as assessing mental health, subjective wellbeing, or job satisfaction (showcased in our team’s prior research), though our focus here is its specific application to gratitude to God. We believe this to be a strength of the paper in that readers can also generalize these principles to other topics, making it more usable and citable.

  1. The authors give several examples of phrases that indicate gratitude to God which are typically found on social media platforms.  These certainly seem to match my subjective impressions of social media, but I wonder if the authors could also make their argument even more compelling by attaching some numbers to the frequency of these behaviors.  Of course a full study of online expressions of gratitude requires a different, more empirically-focused paper, but if the authors could do a quick search for the frequency of certain terms (e.g., “praise God”) it could provide even more support for their claim that this is an important real-world behavior worthy of study.

This is a great point and we have added more information here to provide support for the claim. See below

Our team applied search terms (20 key terms, 10 God focused and 10 interpersonal/general) to find relevant tweets, which resulted in a large quantity of observable behavior of gratitude towards God. Limiting it to the year 2019, we found 1.2 million Tweets referencing gratitude; of the subsample of 105k Tweets that were initially studied, 29.3% of the Tweets referenced gratitude toward God. 

  1. On page 2 the authors argue that “Cultural knowledge is necessary to understand these language norms and variations to properly measure them for cross-cultural studies.”  They could provide further discussion of how one could gather this cultural knowledge.  Does it require preliminary self-report data with pilot samples? Ethnography? Recruiting research assistants from the local context you want to study? Consulting polls of social media users?

These are good questions. To address this query about cultural knowledge, we added some example references (Section 2) on what researchers can do. Some examples include forming a cross-cultural team of researchers (Tam & Milfont, 2020) who are involved in the community, use of qualitative or survey responses to understand context-specific knowledge (Broesch et al., 2020), as well as incorporating mixed methods that allow researchers to measure and compare these cross-cultural differences (Schrauf, 2018). 

  1. In describing the potential issues with ecological validity or representativeness of these expressions, the authors state that “the expressions of gratitude to God on public social media may still hold great value in predicting other types of experiences that may occur exclusively on such platforms” (lines 127.128).  This sentence could use more examples of what other types of platform-based experiences might be theoretically-interesting.

We have added in more information here and in the paper (Section 3) to further expand on this.

The idea is that this is the new reality in which people communicate and experience life and so expressions of gratitude to God measured through social media can predict outcomes such as expressions of well-being or engagement on social media.

  1. The authors repeatedly mention the necessity to find good focal words to count in text-type data.  Is there at least some good, validated, pre-existing dictionary of gratitude-related words?  If so, the authors could direct readers to it here.  Alternatively, the authors could be explicit that such a dictionary does not already exist, and that users would need to make their own.

We have added in more information here and in the paper (Section 1) to further expand on this.

We note that researchers more generally can use existing libraries such as the Linguistic Inquiry and Word Count (LIWC; Pennebaker et al., 2015) to curate a gratitude dictionary such as gratef* gratef*, grati*, thank*, or appreciat* (the “*” symbol means that all words that start with the same letters are included) (e.g., Anicich et al., 2022) and also consider specific gratitude terms (e.g., praying hands emoji, tgif). Such an approach would still need domain specific validation, as dictionaries do not necessarily capture nuances in language. A natural language processing based approach on the other hand is able to capture and learn from linguistic patterns (e.g., sequences of words), thus with the potential to provide better validity to gratitude measurements.

  1. The authors discuss extensively the issues with potential lack of authenticity (e.g., ulterior motives, sarcasm, lack of prototypicality) in gratitude expressions on social media.  This implies that expressions on social media may be limited in their ability to accurately capture internal mental states.  It also seems to imply that this method may be better suited for testing research questions pertaining to witnessingother people’s expressions of gratitude, or why people choose to express gratitude in public forums like social media.  This may be another way that social media analyses could offer insights that are distinct from (not merely convergent with) self-report data.

Thank you for this point. We have added in this insight in our paper.

More generally, expressions of gratitude to God on social media may not accurately capture a person’s internal states, although they may still be very useful for assessing perceptions and reactions of others witnessing these gratitude expressions; or to understand why people express gratitude online publicly. In short, gratitude expressions online and self-reported gratitude may have only some degree of overlap and not have high convergence, but they could have good divergent validities in predicting different outcomes.

  1. A definition could be provided for the term “the digital divide” (line 217).

Yes, we have provided a definition in the paper now at the location noted above.

  1. The authors could reference which specific methods can be used to do the actual text gathering and analysis, or at least reference some papers that provide more thorough and concrete guidelines on these methods.

Thank you, we have provided some references on papers that provide specific use of the method such as using LIWC (Pennebaker et al., 2015).

  1. There are a few typos that can be corrected with further proofreading (e.g., the repeated citations in lines 32/33).

Thank you for noticing. We have now caught the typos.